# `HeQ`: a Large and Diverse Hebrew Reading Comprehension Benchmark

**Amir DN Cohen**[1]    **Hilla Merhav-Fine**[2]    **Yoav Goldberg**[1,3]
**Reut Tsarfaty**[1]

[1]Bar-Ilan University, Ramat-Gan, Israel
[2]Webiks, Tel Aviv, Israel
[3]Allen Institute for AI, Seattle, WA

amirdnc@gmail.com, hilla@webiks.com, yoav.goldberg@gmail.com, reut.tsarfaty@gmail.com

## Abstract

Current benchmarks for Hebrew Natural Language Processing (NLP) focus mainly on morpho-syntactic tasks, neglecting the *semantic* dimension of language understanding. To bridge this gap, we set out to deliver a Hebrew Machine Reading Comprehension (MRC) dataset, where MRC is to be realized as extractive Question Answering. The morphologically-rich nature of Hebrew poses a challenge to this endeavor: the indeterminacy and non-transparency of span boundaries in morphologically complex forms lead to annotation inconsistencies, disagreements, and flaws in standard evaluation metrics. To remedy this, we devise a novel set of guidelines, a controlled crowdsourcing protocol, and revised evaluation metrics, that are suitable for the morphologically rich nature of the language. Our resulting benchmark, `HeQ` (**He**brew **Q**A), features 30,147 diverse question-answer pairs derived from both Hebrew Wikipedia articles and Israeli tech news. Our empirical investigation reveals that standard evaluation metrics such as F1 Scores and Exact Match (EM) are not appropriate for Hebrew (and other MRLs), and we propose a relevant enhancement. In addition, our experiments show low correlation between models' performance on morpho-syntactic tasks and on MRC, which suggests that models that are designed for the former might underperform on semantics-heavy tasks. The development and exploration of `HeQ` illustrate some of the challenges MRLs pose in natural language understanding (NLU), fostering progression towards more and better NLU models for Hebrew and other MRLs.

## 1 Introduction

*Machine reading comprehension* (MRC), a critical skill of NLP systems, is often assessed by means of testing the ability of models to answer questions about a given passage, a setup known in NLP also by the name Question Answering (QA).

QA systems find utility in a large spectrum of applications, from chatbots (Gao et al., 2018) and search engines (Guu et al., 2020) to the evaluation and assessment of pre-trained models' (Devlin et al., 2019; Joshi et al., 2020; Lan et al., 2019) comprehension and reasoning skills. Despite these developments, the landscape of MRC is primarily dominated by datasets and models for English, where low- to medium-resource languages such as Hebrew remain largely underrepresented.

Although recent research in NLP shows increased interest in the Hebrew language (Seker et al., 2022; Guetta et al., 2022), the focus remains largely on morpho-syntactic tasks. With the exception of ParaShoot (Keren and Levy, 2021), the absence of Natural Language Understanding (NLU) benchmarks for Hebrew leads to the design of models primarily tailored for morpho-syntactic tasks. However, it remains an open question whether the performance on existing morpho-syntactic benchmarks correlates with actual natural language understanding, and indeed, as we demonstrate in §5, this is not necessarily so.

A central challenge in the development of QA systems for Hebrew, a Morphologically Rich Language (MRL), compared with morphology-impoverished ones, is the identification of precise answer spans in morphologically rich internally complex surface forms. This is exacerbated by multiple processes of affixation in Hebrew (prefixes, suffixes, clitics) that may obscure the boundaries between linguistic units, making it harder to pinpoint the correct span of answers. For instance, given the sentence אתמול הייתי בבית. (literally: "yesterday I-was at-the-house"), valid responses to the question "Where was I?" can be either בבית (at-the-house), or בית (house). Current annotation guidelines, crowdsourcing protocols and evaluation metrics, however, assume the annotated spans respect space-delimited token boundaries, and fail to accommodate such morphological patterns, conse-

quently penalizing boundary alterations and causing annotation disagreements. To counter this, we propose a revised set of guidelines and new evaluation metric, *Token-Level Normalized Levenshtein Similarity* (TLNLS), which manifests lower sensitivity to affixation changes.

We thus present `HeQ`, a new benchmark for MRC in Hebrew, modeled around the format of the widely-used Stanford Question Answering Dataset (SQuAD) (Rajpurkar et al., 2016a, 2018a). SQuAD is a benchmark dataset for MRC that was collected via crowd-sourcing and consists of questions and answers based on Wikipedia articles. However, SQuAD has been shown to contain biases and spurious regularities that allow models to exploit heuristics and shallow cues without fully understanding the text (Jia and Liang, 2017). Furthermore, SQuAD covers only a limited range of topics and domains (Ruder and Sil, 2021). Hence, in this work, we introduce several novelties. First, we work with a fixed set of human annotators and train each annotator individually based on their previous annotations to increase the quality and diversity of their annotations. We also add a data source from a news domain (GeekTime), besides Wikipedia, to increase `HeQ` genre diversity. Finally, we change the unanswerable questions generation methodology so they become more challenging for the model.

In our experiments, we observe that a multilingual model, mBERT outperforms all other Hebrew-trained models despite being exposed to the least amount of Hebrew data during training, and having the smallest Hebrew token vocabulary. This suggests that pretrained models in Hebrew can greatly benefit from pretraining on other languages. Furthermore, we find that models trained on our new dataset, `HeQ`, perform significantly better than those trained on the earlier ParaShoot dataset, which emphasizes that the quality of the collected data is greatly influential on model performance (in fact, a model trained on a subset of `HeQ` with the same number of questions as in the ParaShoot dataset, outperforms a model trained on ParaShoot, also on the ParaShoot test set). Lastly, models based on the Geektime section outperform the Wikipedia ones and demonstrate better domain transferability, likely due to the former's varied text structure.

In conclusion, despite the complexity of Hebrew, multilingual pretraining proves beneficial for MRC tasks. The quality and diversity of the data appear to be as important as the size, underscoring the importance of dataset design and the value of our new Hebrew MRC benchmark, `HeQ`. Note that all data, tagging platform, user manual, and the training and evaluation scripts are publically online.[1]

## 2 The Challenge: Extractive Question Answering in Morphologically Rich Languages

Extractive Question Answering (Extractive-QA) is a common and widely used task where a model is tasked with locating and reproducing an answer to a question posed directly with regard to a source text. In morphologically-rich languages, this task becomes more complex due to the linguistic features that these languages manifest. Morphologically-rich languages, such as Hebrew, Turkish, or Arabic, exhibit high morphological complexity, including extensive inflection, agglutination, and derivational morphology. This characteristic makes Extractive-QA more challenging because the model needs to process and understand complex word forms and grammatical constructions to correctly pinpoint the answer spans.

In these languages, a single word can carry information typically spread across several words in languages with more impoverished morphology, such as English. This means that word- and sentence-boundaries may not always align with semantic units or logical chunks of information. Consequently, Extractive-QA models must be able to handle this morphological richness to accurately locate and extract the correct answer. Due to the high inflectional nature of these languages, the correct answer may appear in different forms within the text, requiring the model to understand and match these varied forms. This aspect necessitates an understanding of the language's morphological rules and a richly annotated dataset to provide the model with enough examples to learn from.

On top of that, existing evaluation metrics, primarily designed for morphologically simpler languages, often fall short when applied to languages with higher morphological richness. A major contributor to this is affixations, which are a prevalent feature in MRLs. These languages often merge what would be considered separate words in languages like English into a single unit. This means that an answer span might not align with the tradi-

---

[1] https://github.com/NNLP-IL/Hebrew-Question-Answering-Dataset

tional notion of word boundaries.

Current evaluation metrics do not take this into account, leading to potential harsh penalties for minor boundary deviations. Metrics like Exact Match (EM) or F1 score, which function well in morphologically simpler languages, may not accurately reflect a model's performance in morphologically rich languages. For instance, slight changes in word endings might result in a dramatically different EM score, even if the overall answer meaning remains intact. Similarly, an F1 score may fail to capture the nuances in word formation and changes that are inherent to these languages.

## 3 Collection Process

This section provides a comprehensive account of the data preparation, annotation, and validation procedures employed in this study. Specifically, we describe the methods used to preprocess the raw data and prepare it for annotation, the rigorous annotation process carried out to ensure high-quality data, and the validation steps taken to evaluate the reliability and accuracy of the resulting dataset.

### 3.1 Annotation Philosophy

HeQ was created with four principles in mind:

**Diversity.** There is evidence that the quality of the model (in both pre-training and fine-tuning) is based on the diversity of the datasets (Zhou et al., 2023). We increase the diversification of the datasets in three ways: First, we generate the data from two sources, Wikipedia and News, which have very different structures, and topics, which resulted in different question types. The second is the paragraphs chosen for annotation, we made sure that the initial pool of paragraphs was as diverse as possible. And thirdly, annotators were explicitly instructed to create different and diverse sets of questions. We also monitored the diversity of annotations during the questions collection process.

**Accuracy** Accuracy is important for both training robust models and accurate evaluation. We detailed the measurements we took to ensure very high accuracy in §3.3 and present the validation metrics in §3.4.

**Difficulty** SQuAD was criticized to encourage the model to learn shallow heuristics (Elazar et al., 2022). We aimed to overcome this by instructing the annotators to produce questions that require some inference (see "gold" questions in Table 1).

In addition, the annotators were monitored and have been given feedback on the overlap between the questions and the text, aiming to minimize it.

**Quality over quantity.** While HeQ is much smaller than SQuAD (30K vs 150K samples), we argue that the quality, diversity, and difficulty of the data are more important than the sheer number of examples. To demonstrate this, in §5 we compare different sizes of HeQ and compare HeQ to the earlier ParaShoot dataset (Keren and Levy, 2021).

In the following subsections, we describe how we created HeQ based on these principles.

### 3.2 Data Preparation

The HeQ dataset was created by combining two distinct data sources, namely, Hebrew Wikipedia and Geektime, an Israeli technology newspaper. From the Hebrew Wikipedia, we selected the 3,831 most popular articles to create the Wikipedia paragraph pool. Subsequently, we randomly extracted paragraphs from these articles, with a constraint that each paragraph should contain between 500 and 1,600 characters and no more than three paragraphs per article. We removed paragraphs that contained non-verbal sections, such as mathematical equations, and those that could be considered highly sensitive or inappropriate for our dataset, such as those containing violent, graphic, or sexual content. Some paragraphs were manually replaced with alternative paragraphs to ensure a diverse set of topics and styles. After this filtering and manual replacement process, the paragraph pool consisted of 5,850 paragraphs, across a total of 2,523 articles.

To obtain the Geektime paragraph pool, we used articles published from the year 2017 onwards, excluding those in the Career Management category as they contained job descriptions and not news. We randomly selected paragraphs from the remaining pool of articles, with a minimum length of 550 characters and at least 300 characters in Hebrew.

### 3.3 Annotation Process

**Crowed-workers recruitment.** We ensured annotation quality by recruiting native Hebrew speakers, who self-identified as fluent, via the Prolific crowd-sourcing platform. Candidates were provided an annotation guide, outlining task requirements and guidelines for question creation and answer identification.

The selection process involved a trial task, where candidates crafted questions and identified answers

from given paragraphs. Their performance was meticulously evaluated by the authors on parameters like correct answer span, question-answer lexical overlap, and attention to detail, ensuring the exclusion of frequent typographical errors, poorly articulated questions, or incomplete answers. Only candidates demonstrating comprehensive task understanding and the ability to deliver high-quality annotations were recruited.

**The Annotation Process** For the annotation, we used a custom UI, based on the UI used in ParaShoot. where in each step the annotator is presented with a paragraph and is required to write a question and an answer, and mark if the question is answerable. We show the UI in Figure 1 under Appendix A. For each paragraph, the annotators were instructed to create 3-5 questions.

**Crowdsource Monitoring.** During the annotation process, we ensured ongoing quality control over the work of our annotators by providing personalized feedback to each individual as needed. This feedback highlighted areas for improvement and areas of strength. Annotators who successfully applied the feedback and demonstrated improvement were invited to participate in additional annotation tasks, while those who did not improve were disqualified. Initially, feedback was given to each annotator after they completed their task. However, as the annotation process progressed, experienced annotators were randomly selected for feedback to allow for closer monitoring of new annotators and greater freedom for trusted annotators. For most annotators, We observed a significant improvement in question quality after several rounds of feedback.

To assess the quality of the questions, we assigned each question a score based on four levels: rejected, verified, good, and gold. Questions that did not meet the minimum criteria for the dataset, typically due to semantic or morpho-syntactic errors, unclear questions, or questions that were classified "unanswerable" but were answerable to some extent, were classified as "rejected". "Verified" questions passed the minimum threshold but were relatively straightforward or used similar wording to the relevant sentence in the paragraph. "Good" questions had distinct wording, either lexically or syntactically, from the relevant sentence in the paragraph. Finally, "gold" questions required inference. Table 1 presents examples of each question type.

The dataset was randomly split into train (90%),

development (5%), and test (5%) sets, with the constraint that questions related to a specific article were assigned to the same set. Each sample in the test and train was augmented by creating several correct answer spans for better validation.

### 3.4 Data Assessment

To ensure the quality of the evaluation, we manually validated the test and development sets and corrected wrong answers where necessary. Overall, we reviewed 34% of the data, with 6.36% being rejected due to quality concerns. However, this rejection rate was largely attributed to disqualified annotators and does not reflect the overall quality of the dataset as shown in the following paragraph. Also, note that the full development and test sets were validated for correction.

To validate the final dataset quality, we randomly sampled 100 questions from Wikipedia and 100 questions from Geektime for accuracy testing. We measured the quality of questions according to two parameters: the percentage of correct questions (93% for Wikipedia and 97% for Geektime) and the percentage of answers with a correct answer span (100% for Wikipedia and 99% for Geektime).

We include more analysis on the quality of the dataset in Appendix C.

## 4 MRC evaluation for MRLs

Evaluating the performance of MRC models is a fundamental aspect of their development and refinement. However, this evaluation process isn't as straightforward as it might seem — especially when it comes to MRLs. The traditional evaluation metric of MRC is $F_1$ over answer span words (described in Appendix B.1) or a variation thereof. The $F_1$ score is sensitive to small changes in the answer span boundary. For example, if the gold span is "in the house" while the extracted span is "house" the $F_1$ score will be $\frac{2}{3}$. English somewhat alleviates this problem by removing the words "a", "an", and "the" for the evaluation, which in the former example will increase the $F_1$ to $0.5$.

Unfortunately, the boundary issues in MRLs, and in Hebrew specifically, are much more common and result in biased scores. For example, if we consider the same example, but in Hebrew, the phrase "in the house" is translated to the single word בבית, while the word בית corresponds to "house". So while the English phrase will get an $F_1$ score of 0.5, the Hebrew span using a token-based

| Quality label | Example |
|---|---|
| Verified | **Question:**היכן תשודר התכנית קרובים קרובים?
**Sentence:** תוכניות בשידור חוזר המזוהות עם הטלוויזיה החינוכית, כגון קרובים קרובים וזהו זה, תשודרנה בכאן 11.
**Answer:** בכאן 11 |
| Good | **Question:**מי החליף את חאג' מוחמד בתפקידו?
**Sentence:** בפברואר 1939 הסכימה "הוועדה המרכזית" להכיר בחאג' מוחמד כמפקד הכללי של המורדים [...] לאחר מותו מינתה הוועדה המרכזית לתפקיד המפקד הכללי את אחמד מוחמד חסן ("אבו בכר"), מורה לשעבר בכפר בורקה ליד שכם.
**Answer:** אחמד מוחמד חסן |
| Gold | **Question:**מי הבקיע את הגול הראשון בגמר הגביע של 1932?
**Sentence:** אף על פי כן, הצליחה הקבוצה להעפיל בשנת 1932 לגמר הגביע [...] במהלך המשחק במצב של 0–1 להפועל משער של יונה שטרן [...] ננגב הגביע על ידי אחד משחקני הקבוצה.
**Answer:** יונה שטרן |

Table 1: Example for quality levels of different samples.[2]

or word-based evaluation will be scored 0, leading to an underestimation of model performance. This is the result of *compounding*, combining two or more words into a single word in Hebrew. Compounding is very common in Hebrew and happens in many situations like possessive, Pluralization, definite article, prepositions, and other affixations. While some of these affixations also appear in English (like Pluralization), they are much more common in the Hebrew setting.

One way to avoid the issue is to do a complete morphological decomposition of each word in the predicted span and the gold span and remove affixations from the evaluation. Unfortunately, the morphologically rich forms are ambiguous, and Hebrew morphological disambiguation is context-dependent, requiring a dedicated model to perform. We seek an evaluation method that is not dependent on the existence of a disambiguation model and which will not be influenced by mistakes of different disambiguation models. Rather want a method that is simple, unbiased, and easy to run on multiple MRLs. We thus present a novel evaluation metric for MRC tailor-made for MRLs.

### 4.1 Qualities for Evaluation Metric for MRL

We propose a new evaluation metric that takes into account the unique characteristics of MRLs. This metric should have the following characteristics:
∘ **Inflection invariance.** The metric should give a similar score to a word and its inflected and affixed variants (e.g., בית, בבית and הבית).
∘ **Span invariance.** The metric should give a similar score to different spans that represent the same answer ("king David" vs "David").
∘ **language independence.** We aim to create a general evaluation metric for MRLs, that does not require specific knowledge of the target language.

∘ **Speed.** The metric should have a short run time.

### 4.2 The Token-Level Normalized Levenshtien Similarity (TLNLS) Metric

**The Metric.** A candidate for MRC evaluation is the Normalized Levenshtien Similarity metric (which we describe in Appendix B.1.1). This is a character-level metric that has the benefit of being less affected by a change in affixation but causes the evaluation to be skewed when dealing with long words and long sequences, which is undesirable. To overcome this, we present a hybrid metric derived from $F_1$ and Levenshtien similarity. Given two *tokenized* spans, a predicted span $P = \{p_1, \ldots, p_n\}$, and a gold span $G = \{g_1, \ldots, g_n\}$, where $p_i$ and $g_i$ are tokens, we define TLNLS as:

$$\text{TLNLS}(P, G) = \frac{1}{max(|G|, |P|)} \sum_{g_i \in G} \max_{p_i \in P}(ls(g_i, p_i)). \quad (1)$$

**Tokenization.** TLNLS assumes tokenization of the input phrases. There are several ways to tokenize a phrase, using white spaces, using a parser (like YAP (More et al., 2019)), and using a pre-trained model tokenizer. In this work we use white spaces for TLNLS, for several reasons: first, this is a language-independent approach, that can be used for any MRL without any adjustment. Second, the design of TLNLS is such that words with similar stems, but different infliction get a high score. third, any more advanced tokenization might be biased towards a specific model.

**Dates and numbers.** A limitation of these approach concerns dates. E.g. the spans "1948" and "1921" will result in $F_1$ score of 0, but a TLNLS score of 0.5. For this reason in the number of digits

| Type | $F_1$ | Edit | TLNLS |
|------|-------|------|-------|
| Positive | 0.576 | 0.389 | 0.727 |
| Negative | 0.019 | 0.233 | 0.093 |

Table 2: Performance Metrics. The value in the brackets are without dates.

in either span is more than half, we revert to the original $F_1$.

### 4.3 Evaluating the Metric

**Quantitative Evaluation** We evaluate the score the metric assigns to correct spans (positive evaluation) and wrong spans (negative evaluation). Our development set in each case contains several correct spans for each sample.

The positive evaluation is done by taking all the samples in the development dataset that have more than one answer span, and evaluating the similarity between these spans based on each metric. For example, if a sample contains the spans "הבית", "בבית", and "בית", we compare each span to the others (In our example, three comparisons, one for each pair) and report the mean similarity for each pair based on each of the metrics. We assume that a better metric will give a high score for similar spans.

In the negative evaluation, we collect negative spans and validate that the metric gives a low score to these spans compared to the gold answers. To collect these spans we use the trained QA model on the development set and collect answer spans that received very low $F_1$ scores ($< 0.1$) compared to the gold spans, we manually verify that these spans are incorrect. The result is 100 spans that contained a wrong prediction of the model. We compare the model (wrong) output to the gold labels using different metrics.

**Results.** As shown in Table 2, TLNLS achieves the highest score with a large margin in the positive evaluation. The currently used $F_1$ metric achieves a mediocre score of 0.56. In the negative evaluation, $F_1$ achieves the best (lowest) score, while TLNL achieves a slightly worse score of 0.093.

**Qualitative Evaluation** We measure the samples with the greatest score difference between $F_1$ and TLNLS. We iterate over the development set and calculate for each sample the $F_1$ and TLNLS scores (like in the positive evaluation). We show 4 random samples in Table 3. All the samples are underevaluated by $F_1$, (receive a score of 0) even though the returned span is valid.

| Span 1 | Span 2 | TLNLS | $F_1$ |
|--------|--------|-------|-------|
| MusicaNeto-ב | MusicaNeto | 0.909 | 0 |
| לסלבריטאים | סלבריטאים | 0.9 | 0 |
| המוזיאון | מוזיאון | 0.875 | 0 |
| ביתינו | בית | 0.5 | 0 |

Table 3: Qualitative comparison of TLNLS and $F_1$. We sampled 4 examples that had a high gap between $F_1$ and TLNLS.

| Model | EM | $F_1$ | TLNLS |
|-------|-----|-------|-------|
| Aleph Bert | 57.91 | 67.66 | 76.07 |
| Aleph Bert Gimel | 57.12 | 67.37 | 75.3 |
| mBERT | **62.7** | **71.42** | **78.2** |
| mBERT (ParaShoot) | 36.77 | 51.08 | 59.20 |
| ChatGPT | 9.38 | 32.19 | 37.91 |

Table 4: Performance Comparison of Different Models on the HeQ Dataset. All results except Aleph Bert and Aleph Bert Gimel are sgnificant.

### 4.4 Recommended Metric

A good metric is vital for the correct evaluation of different models. Our experiments show that the $F_1$ is too strict in the sense that it gives a low score to good spans (shown by our positive evaluation), including spans that differ only by a single affix. To compensate for this, we suggest a new metric TLNLS, which gives significantly better results for good spans while still giving low scores to bad spans. For the evaluation, we use the two traditional metrics ($F_1$ and exact match), and the new TLNLS metric, which we offer for general use.

## 5 Experiments

We experiment with HeQ in several scenarios. First, we create a baseline based on several known models, then we evaluate the improvement of HeQ trained model vs ParaShoot trained model. We continue by analyzing the impact of the data size of the train data on the models' performance. Finally, we assess the effect of the different domains in the dataset on the overall performance and the cross-domain performance on HeQ. We discuss the training setup in Appendix E

**Improvement compared to ParaShoot.** In table 5 we evaluate the test set of ParaShoot using the ParaShoot train data compared to theh HeQ train data. On all models, we see a significant improvement when training of HeQ compared to ParaShoot.

### 5.1 Baseline Results

We evaluate several known pre-trained models on the HeQ test set. Unless stated otherwise, all mod-

| Model | ParaShoot | | | HeQ | | |
|---|---|---|---|---|---|---|
| | EM | $F_1$ | TLNLS | EM | $F_1$ | TLNLS |
| AB | 26 | 49.6 | 60.41 | 39.23 | 62.42 | 74.73 |
| ABG | 24.68 | 48.87 | 59.8 | 38.96 | 61.92 | 74.11 |
| mBERT | **32.08** | **56.25** | **66.83** | **44.29** | **65.58** | **75.38** |

Table 5: Performance Comparison of Models trained on the ParaShoot and the HeQ Datasets and evaluated on the Parashoot dataset.

| Trainset | EM | F1 | TLNLS |
|---|---|---|---|
| Geektime | 58.38 | 68.00 | 75.80 |
| Wikipedia | 56.05 | 65.72 | 73.47 |
| Geektime + Wikipedia | 62.70 | 71.42 | 78.21 |
| Geektime + Wikipedia (15K) | 59.24 | 67.52 | 75.27 |
| Geektime + Wikipedia (1.7K) | 48.20 | 59.26 | 67.83 |

Table 6: Performance Comparison of Different Training Sets (sizes and domains) on HeQ.

els were trained on the HeQ train set.
○ **AlephBERT (AB)** ([Seker et al., 2022](#)): A Hebrew pre-trained model that is currently used as a baseline for most Hebrew-related tasks.
○ **AlephBERTGimel (ABG)** ([Guetta et al., 2022](#)): A variation on AB which has a larger vocabulary size which resulted in fewer splits and improved performance on the AB evaluation benchmarks.
○ **mBERT** ([Devlin et al., 2019](#)): A multilingual variation of the known BERT model.
○ **mBERT (ParaShoot)** ([Keren and Levy, 2021](#)) - mBERT trained on the parashot dataset.
○ **chatGPT** : ChatGPT is a well-known conversational model that was finetuned on a large LLM from openAI. We evaluate chatGPT using the following prompt: "Answer the following question based on the provided context. If the answer is present in the text, please provide it as a span. If the answer does not exist in the text, reply with an empty string. Question: <question> Context: <context>".[3]

We present the results in Table 4. Similarly to ParaShoot, the best-performing model on all metrics is mBERT, which is surprising considering that mBERT is the pre-trained model that was exposed to the least amount of Hebrew texts during training. We attribute this result to the fact that during the multilingual pretraining, mBERT is the model that was exposed to the largest corpora overall. Based on this hypothesis, future Hebrew pre-trained models might benefit from jointly pre-training on English or other high-resource languages.

---

[3]At the time of writing this paper we could not access the GPT4 API.

Between the Hebrew-monolingual models (AB and ABG), the results are almost comparable, giving AB a small insignificant advantage over ABG. Based on this result we suspect that while the increased vocabulary size has a positive effect on morpho-syntactic tasks like the ones AB and ABG were evaluated on, it has lower significance in semantic tasks like MRC.

Next, the mBERT (ParaShoot) model is underperforming compared to the HeQ trained model. which is not surprising considering the size of HeQ compared to ParaShoot and has better quality.

Surprisingly, the worst model in our evaluation is chatGPT. While it is the largest model evaluated, it achieves the lowest results, the returned spans are often mistaken, and even when the model returns the correct answer the span boundaries are wrong.

## 5.2 The Effect of Data Size and Type on Model Performance

In this subsection, we examine the effect of the number of samples, and the type of data, on the overall performance of the trained models. The results are presented in Table 6.

First, we consider three splits of the data: Geektime (news) only, Wikipedia only, and a subset of the dataset that is roughly the same size as the two other splits (Geektime + Wikipedia 15K). All models were evaluated on the HeQ test set. The Geektime dataset outperforms Wikipedia and the combined split. We hypothesize that the reason for that is that while Wikipedia has more diverse topics, its text follows a relatively rigid structure.

The second experiment is evaluating the model performance based on different data sizes. We compare three data sizes, the full dataset (Geektime + Wikipedia), the 15K version, and 1.7K version, which is comparable in size to ParaShoot. The bigger the dataset is the more performance improves, but interestingly, moving from 1.7K to the full version ($\times 20$ increase) increases the final score by only 11.4% TLNLS points, and moving from 15K to the full version ($\times 2$ increase) increases the final score by only 3% TLNLS points. Based on this we assume that to improve future models a 100K+ dataset is needed. In addition, the large gap between the 1.7K version and ParaShoot (67.83% to 59.2% TLNLS) shows that most of the performance comes from high-quality and challenging questions and not from the sheer size of the dataset.

| Split | Geektime | | | Wikipedia | | |
|---|---|---|---|---|---|---|
| | EM | $F_1$ | TLNLS | EM | $F_1$ | TLNLS |
| Geektime | 61.07 | 70.59 | 77.35 | 55.70 | 65.36 | 74.36 |
| Wikipedia | 52.20 | 63.39 | 71.14 | 59.81 | 68.04 | 75.65 |
| Geektime + Wikipedia (15K) | 59.33 | 68.65 | 76.33 | 59.15 | 66.38 | 74.09 |
| Geektime + Wikipedia | 64.40 | 73.36 | 79.98 | 60.74 | 69.46 | 76.92 |

Table 7: Performance Comparison of Different Trainset on Geektime and Wikipedia Datasets

## 5.3 Domain Transfer

HeQ contains two domains: Geektime (tech news) and Wikipedia. We evaluate the models in both in-domain and cross-domain settings. We also add Geektime + Wikipedia (15K) and the full dataset (Geektime + Wikipedia) as baselines.

The results are presented in Table 7. Similarly to §5.2, we see that the Geektime-trained models achieve better results than the Wikipedia one, having a 6.21 points increase in TLNLS on the Geektime test, while the Wikipedia train model has only 1.29 points TLNLS increase.

We again attribute this to the similarity in text structure in Wikipedia compared to the less strict text structure in Geektime.

## 6 Related Work

MRC datasets (often these take the form of question-answering (QA) tasks) have been developed by the NLP community for a long time (Hirschman et al., 1999). While newer datasets are being developed, the structure of QA data — a paragraph (from Wikipedia), a question about this paragraph, and an answer — is still the way most datasets are constructed nowadays. Perhaps the most well-known datasets are the SQuAD and SQuAD 2.0 (Rajpurkar et al., 2016b, 2018b) which use Wikipedia as a source for paragraphs, creating 100,000 QA instances. SQuAD 2.0 improve over the first version by including another 50,000 unanswerable questions.

Other known datasets are NewsQA (Trischler et al., 2017) that were collected from news articles, TriviaQA (Joshi et al., 2017), a large-scale dataset (650K samples) that used questions from quiz web sites aligned with Wikipedia and web results, and CoQA (Reddy et al., 2019), a dataset focused on conversational question answering, offers a unique context-based approach with dialogues and answers. Another recent QA dataset is natural questions (Kwiatkowski et al., 2019), which distinctly uses natural queries from Google searches, aligned with Wikipedia paragraphs.

All of the above are English datasets. In contrast, TyDi QA (Clark et al., 2020) is designed for multilingual information-seeking QA, encompassing languages from various language types, offering a more global scope for MRC research.

While the selection of MRC datasets in English is large, the only MRC dataset in Hebrew is ParaShoot (Keren and Levy, 2021), consisting of Wikipedia articles and constructed in a similar way to SQuAD. We improve over ParaShoot in three aspects: size, diversity and quality. HeQ has 30K questions compared to 3K; uses news and Wikipedia domains; and was much more rigorously validated. In particular, special attention has been given to the evaluation of flexible span boundaries.

While many MRC datasets leverage external sources (user queries and trivia sites), these resources are not available for Hebrew. Therefore, we create HeQ in a similar fashion to SQuAD 2.0. However, revised guidelines have been introduced in order to ensure diversity, question difficulty, and sorting our matters concerning span boundaries.

## 7 Conclusion

In this work, we presented the challenges of MRC for MRLs. We introduced HeQ, a Hebrew MRC benchmark designed to enable further research into Hebrew NLU. Experimental insights highlighted the surprising efficacy of mBERT, trained on diverse languages, over Hebrew-monolingual models, suggesting the value of multilingual pretraining.

However, data quality was seen to be as important as size, underlining the need for high-quality, diverse Hebrew datasets. The complexity of evaluating models in MRLs was also recognized, leading to the proposal of the TLNLS metric, better suited to handle morphological variations.

Overall, this work advances the understanding of MRC in morphologically rich languages, offering avenues for improved benchmarks, datasets, and evaluation metrics. It invites further exploration to broaden knowledge and enhance MRC performance in underrepresented languages.

## 8 Limitation

We recognize two main limitations of the dataset:

**Dataset Size.** Although `HeQ` is of high quality, it consists of a relatively smaller number of instances compared to well-established English MRC datasets like SQuAD. The limited dataset size might impact the model's ability to generalize and capture the full range of linguistic and contextual variations in Hebrew. However, efforts have been made to ensure the dataset's quality and diversity compensates for its smaller scale.

**Lack of Natural Questions.** One of the limitations faced in constructing `HeQ` is the unavailability of a dedicated resource containing natural questions in Hebrew. Natural questions, which reflect authentic and unconstrained human language use, provide a realistic and diverse set of challenges for MRC models. Without access to such a resource, the dataset may not fully capture the breadth and complexity of real-world questions, potentially limiting the evaluation of MRC models' performance in natural language understanding.

Despite these limitations, the introduction of a new dataset and evaluation metric provided in this paper are valuable contributions to the field and the progression of Hebrew NLP research.

## 9 Ethics and Broader Impact

This paper is submitted in the wake of a tragic terrorist attack perpetrated by Hamas, which has left our nation profoundly devastated. On October 7, 2023, thousands of Palestinian terrorists infiltrated the Israeli border, launching a brutal assault on 22 Israeli villages. They methodically moved from home to home brutally torturing and murdering more than a thousand innocent lives, spanning from infants to the elderly. In addition to this horrifying loss of life, hundreds of civilians were abducted and taken to Gaza. The families of these abductees have been left in agonizing uncertainty, as no information, not even the status of their loved ones, has been disclosed by Hamas.

The heinous acts committed during this attack, which include acts such as shootings, sexual assaults, burnings, and beheadings, are beyond any justification.

We fervently call for the immediate release of all those who have been taken hostage and urge the academic community to unite in condemnation of these unspeakable atrocities committed by Hamas, who claim to be acting in the name of the Palestinian people. We call all to join us in advocating for the prompt and safe return of the abductees, as we stand together in the pursuit of justice and peace.

This paper was finalized in the wake of these events, under great stress while we grieve and mourn. It may contain subtle errors.

## Acknowledgements

We thank Roei Shlezinger and Tal Geva for their invaluable contributions and assistance in shaping this paper.

We thank the anonymous reviewers for their insightful comments and fruitful discussions.

This last author was funded by the Israeli Ministry of Science and Technology (MOST grant number 3-17992), and an Israeli Innovation Authority grant (IIA KAMIN grant), for which we are grateful. In addition, This project has received funding from the European Research Council (ERC) under the European Union's Horizon 2020 research and innovation programme, grant agreement No. 802774 (iEXTRACT).

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

# A  Annotation UI

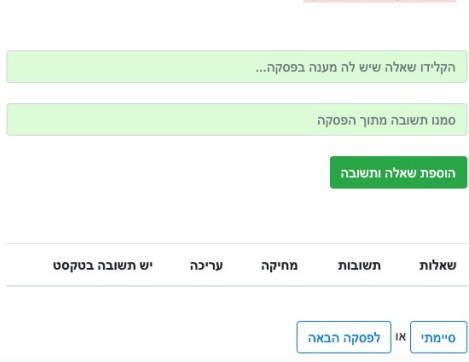

Figure 1: Annotation UI.

# B  Metrics

## B.1  $F_1$ Score

The current metric used for many MRC datasets evaluation is the $F_1$ score, defined as follows.

Given a stream of tokens for the gold span $G = \{g_1, \ldots, g_n\}$ and a stream of tokens for the predicted span $P = \{p_1, \ldots, p_n\}$. we define the intersection function as

$$\text{intersect}(G, P) = $$
$$|\{g_i : \exists\ p_i \text{ s.t. } p_i = g_i, g_i \in G, p_i \in P\}|.$$

Using this we can define the known precision and recall, and F1 functions:

$$\text{precision}(G, P) = \text{intersect}(G, P)/|G|.$$

$$\text{recall}(G, P) = \text{intersect}(G, P)/|P|.$$

$$F_1(G, P) = \frac{2 \cdot \text{recall}(G, P) \cdot \text{precision}(G, P)}{\text{recall}(G, P) + \text{precision}(G, P)}.$$

### B.1.1  Normalized Levenshtien Similarity

Levenshtien distance is defined as the minimal number of edits that are required to convert a string $s_1$ into a different string $s_2$ (Navarro, 2001). An edit is one of the following functions:

- **Removal** - remove a character from a string (e.g. "cat" → "ca").

- **Addition** - add a new character to a string (e.g. "cat" → "cats").

- **Substitution** - change a character from a string to a different character (e.g. "cat" → "cut").

Formally we can define Levenshtien distance recursively as:

$$lev(s_1, s_2) =$$
$$\begin{cases} |s_1|, & \text{if } |s_2| = 0 \\ |s_2|, & \text{if } |s_1| = 0 \\ lev(s_1, s_2), & \text{if } s_1[0] = s_2[0] \\ 1+ \\ \min \begin{cases} lev(s_1[1:], s_2) \\ lev(s_1, s_2[1:]) \\ lev(s_1[1:], s_2[1:]), \end{cases} & \text{otherwise.} \end{cases}$$

Where the "[]" operator is the array operator in Python. To normalize the edit distance we simply divide the result by the max length of the two strings. To use normalized Levenshtien *similarity* we use the following formula:

$$\text{ls}(s_1, s_2) = 1 - \frac{lev(s_1, s_2)}{max(|s_1|, |s_2|)}.$$

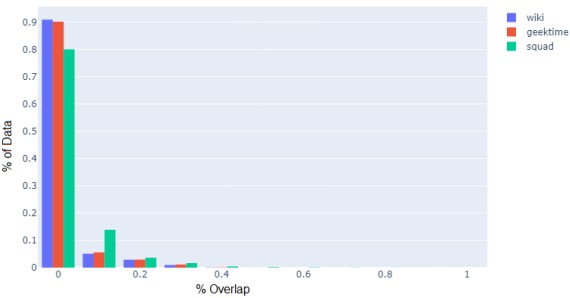

(a) Question and answer overlap.

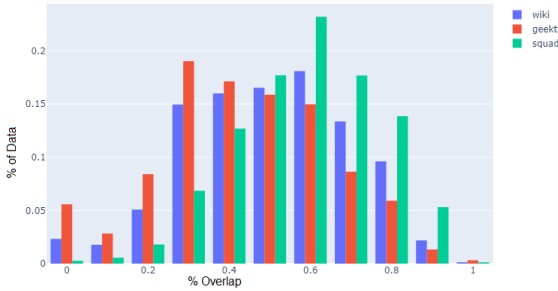

(b) Question and context overlap

Figure 2: A comparison of the overlap in `HeQ`. on both metrics, our dataset has less overlap compared to SQuAD.

| Parameter | Value |
|---|---|
| Learning rate | $3 \cdot 10^{-5}$ |
| batch size per device | 4 |
| GPU type | four Nvidia 2080TI |
| Optimizer | Adam |
| Epochs | 5 (with early stopping) |

Table 8: Training parameters.

## C Empirical Analysis

In this section, we supply more analysis driven by other studies on MRC datasets.

**Data Overlaps.** Biases in the data result in a model that learns the biases and not the actual task (Elazar et al., 2022; Rondeau and Hazen, 2018). We aimed to reduce biases as much as possible. In this analysis, we validate that there is little overlap between the question and the text and between the question and the answer (in answerable samples). We plot the overlap between the questions and the context and answer in 2. It is easy to see that `HeQ` has a lower overlap compared to the SQuAD dataset.

## D English Translation of Example from the Data

We show the translation of Table 1 in Table 9

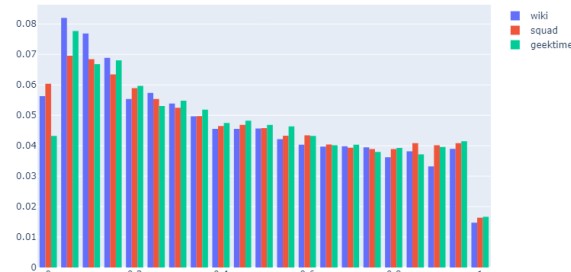

Figure 3: The distribution of answers inside the context on `HeQ` and SQuAD. The distribution is roughly the same.

**Position Bias.** Uneven distribution of the answer location might also lead to biases in the data (Ko et al., 2020; Shinoda et al., 2022). We show the answer distribution across the context in Figure 3. Like SQuAD, the distribution of answers in `HeQ` is mostly uniform, where there is a bias towards the start of the context and a negative bias towards its end.

## E Training Parameters

Table 8 shows the hyper-parameters used for training. No hyper-parameter tuning was made during our experiments.

| Quality label | Example |
|---|---|
| Verified | **Question**:Where will the program be broadcast in the coming days?> 
 **Sentence**: Rebroadcast programs associated with educational television such as "Close Relatives" and "Zeho ze" will be broadcast on kan 11 
 **Answer**: on kan 11 |
| Good | **Question**: Who replaced Hajj Muhammad in his position? 
 **Sentence**: In February 1931, the "Central Committee" agreed to recognize Hajj Muhammad as the general commander of the rebels [...] After his death, the Central Committee appointed Ahmed Muhammad Hassan ("Abu Bakr"), a former teacher in the village of Burka near Nablus, to the position of general commander. 
 **Answer**: Ahmed Muhammad Hassan |
| Gold | **Question**: Who scored the first goal in the cup final of 1932? 
 **Sentence**: Nevertheless, the team managed to qualify in 1932 for the cup final [...] during the game in a 1-0 situation to Hapoel from a goal by Yona Stern [...] the trophy was stolen by one of the team's players 
 **Answer**: Yona Stern |

Table 9: Example for quality levels of different samples in English.