# OpenReview forum: "HeQ: a Large and Diverse Hebrew Reading Comprehension Benchmark"
_EMNLP/2023/Conference — EMNLP 2023 Findings_

### Official Review · Reviewer_ujfx · 2023-08-01

**Soundness:** 3

**Excitement:**

4: Strong: This paper deepens the understanding of some phenomenon or lowers the barriers to an existing research direction.

**Missing References:**

Line 122: missing reference for paraShoot dataset
Line 274-275: any website link for the Prolific crowd-sourcing platform?


**Paper Topic And Main Contributions:**

The paper proposes a new Hebrew dataset, HeQ, for the Machine Reading Comprehension task. The dataset is created by following a set of guidelines and a crowdsourcing protocol to control the quality of it. Compared with the counterpart (ParaShoot), HeQ includes one more domain of news showing the diversity of it. Due to Hebrew is a morphological-rich language, the authors propose a hybrid evaluation metric, TLNLS, derived from F1 and Levenshtien similarity. Thus, the models can be evaluated fairly considering flexible span boundaries. Finally, the authors conduct experiments from different perspectives, performance of baselines on the datasets, data size and domains, to show the effectiveness and contributions of the benchmark presented in this work.

**Questions For The Authors:**

In lines 342-343, I am curious that if the authors will delete all annotations of those disqualified annotators after rejecting the cases they made.

**Reasons To Accept:**

The proposed benchmark in low-resource languages is quite valuable for the Machine Reading Comprehension (MRC) community to foster further research on it. It is quite a desired work. The motivation and idea of the work are clear and convincible. Most of claims and arguments are supported well with relevant literature. The writing is good. Overall, it is a solid work and makes meaningful contributions in the MRC field.



**Reasons To Reject:**

I doubt the generalisation ability of proposed TLNLS metric considering flexible span boundaries in other morphological-rich languages. The reason of languages that have complexity in morphology is different. Prefixes or suffixes could change meaning of words totally although they have same stems.  In Arabic, different forms of words could present same semantic meaning, and same word also could be presented in various forms. The work only applies TLNLS in Hebrew. It is still need to be investigated showing the ability of TLNLS metric in other morphological-rich languages.

The example in Table 1 without translation in English is not understandable.

In line 434, what does ls denote in the formula? It is better give a brief explanation of it.

The caption of Table 4 is not very clear to me.


**Reproducibility:**

4: Could mostly reproduce the results, but there may be some variation because of sample variance or minor variations in their interpretation of the protocol or method.

**Reviewer Confidence:**

3: Pretty sure, but there's a chance I missed something. Although I have a good feel for this area in general, I did not carefully check the paper's details, e.g., the math, experimental design, or novelty.

**Typos Grammar Style And Presentation Improvements:**

I don't know what MRL stands for in line 25 of abstract when I read there.

What does 'Edit' means in Table2?

Line 525: typos in 'dname'.

---

> ### Author Rebuttal · Authors · 2023-08-28
>
> Thank you for your review! We were glad to read that you found our work to be solid and a meaningful contribution to the MRC field!
> Regarding your question “In lines 342-343, I am curious if the authors will delete all annotations of those disqualified annotators after rejecting the cases they made.”-
> To make this clear: we removed all wrong annotations from disqualified annotators, they are not part of the final dataset. We will clarify it in the camera ready.
>
> Regarding your reason to reject
> Reason1: “I doubt the generalization ability of the proposed TLNLS metric considering flexible span boundaries in other morphological-rich languages. The reason for languages that have complexity in morphology is different. Prefixes or suffixes could change the meaning of words totally although they have the same stems. In Arabic, different forms of words could present same semantic meaning, and same word also could be presented in various forms. The work only applies TLNLS in Hebrew. It is still need to be investigated showing the ability of TLNLS metric in other morphological-rich languages.”
>
> Answer: The purpose of TLNLS is to be a good discriminator between answers of models given the same question and the same text and not between two random spans (we will clarify this point in the paper). Hebrew, like Arabic, exhibits both types of morphemes: the first type includes affixes that are attached to base words to indicate grammatical information, such as inflectional morphemes, along with prefixes that function as conjunctions, prepositions or definite articles. The second type involves derivational morphemes, which are affixes that are added to a base word to create a new word with a different meaning or part of speech. The reason that we focus on the former category is that they are a lot more common as mistakes done by the model. In fact, after reading your comment we searched the data for mistakes that involved derivational morpheme, and didn’t find any. So, we argue that TLNLS can be used effectively for other MRLs, and especially Arabic, as its grammar and inflections are relatively similar to Arabic.
> Following your review we also applied  TLNLS on an Arabic MRC dataset ( GitHub - RanaMalhas/QRCD: Reading comprehension on the Holy Qur'an ). Based on our test, TLNLS is also very relevant for this setup, but we believe that the full analysis of TLNLS in Arabic is beyond the scope of this current paper.
>
>
> Reason2: “The example in Table 1 without translation in English is not understandable.”
> Answer: We include the translation in the appendix, on Table 9, and following your comment, we will add it to Table 1 for camera ready.
>
> Reason3: In line 434, what does ls denote in the formula? It is better to give a brief explanation of it.
> Answer: We will include a full explanation in the camera-ready version. The idea behind this equation is to split the span into words, and then we score each word based on its similarity using normalized Levenshtein similarity. Note that ls - levenshtein similarity is explained in Appendix B, we will move this notation to the relevant section in the camera ready.
>
> Reason4: The caption of Table 4 is not very clear to me.
> Answer: We will make the caption clearer, specifically we’ll explain how we calculated significance.
>
> Regarding typos and references - we will fix all of them for the next version.

---

### Official Review · Reviewer_mCKP · 2023-08-05

**Soundness:** 3

**Excitement:**

3: Ambivalent: It has merits (e.g., it reports state-of-the-art results, the idea is nice), but there are key weaknesses (e.g., it describes incremental work), and it can significantly benefit from another round of revision. However, I won't object to accepting it if my co-reviewers champion it.

**Paper Topic And Main Contributions:**

This paper introduces a machine reading comprehension dataset for the Hebrew language in the form of a QA dataset (HeQ). The authors demonstrate issues with the usual metrics for evaluating QA performance in Hebrew and propose a new metric that is a modified version of Normalized Levenshtein Similarity. Finally, the authors evaluate several models on HeQ, finding that mBERT performs the best and even outperforms ChatGPT.

**Questions For The Authors:**

QA: In 3.1 (Annotation Philosophy), you mention that you increased diversification in three ways. The first was clear: include different structures and different topics. The other two methods just say that you made sure the dataset was “diverse”—what metrics are you using for diversity?
QB: How did you create the multiple answer spans to augment the train and test samples?
QC: In line 351, what is a “correct question?”
QD: At the beginning of 4.3, do you just mean “pairwise?”
QE: In line 452, did you mean if the number of digits in either spam is more than half the total number of digits in each span?


**Reasons To Accept:**

This dataset is a resource for which no alternatives currently exist—the authors motivate the need for this resource well by pointing out unique aspects of the Hebrew language.

The dataset quality appears to be very high. The authors clearly detailed their data curation attempts and manually verified much of the data.

The domain transfer and data size experiments are particularly compelling.

**Reasons To Reject:**

The proposed evaluation metric (token-level normalized Levenshtein Distance) is not validated by humans. The other metrics that are reported in the paper (F1 and exact match) are shown by the authors to be ineffective. This makes it difficult to trust the assessment of model performance on HeQ.

**Reproducibility:**

5: Could easily reproduce the results.

**Reviewer Confidence:**

3: Pretty sure, but there's a chance I missed something. Although I have a good feel for this area in general, I did not carefully check the paper's details, e.g., the math, experimental design, or novelty.

**Typos Grammar Style And Presentation Improvements:**

There are many small typos—I would give the entire paper a reread. The ones I caught:

line 522 “on” not “of”
Crowd workers spelled wrong twice in subheadings in 3.3
Write out MRL in abstract
Line 525
Lines 448 and 451

---

> ### Author Rebuttal · Authors · 2023-08-28
>
> Thank you for the review! we were glad to read you found our work well-motivated, useful, and the dataset of high quality -- this precisely what we aimed for in constructing this resource.
>
> Regarding the evaluation (reason to reject):
> "The proposed evaluation metric (token-level normalized Levenshtein Distance) is not validated by humans. The other metrics that are reported in the paper (F1 and exact match) are shown by the authors to be ineffective. This makes it difficult to trust the assessment of model performance on HeQ."
>
> Thank you for bringing up the critical aspect of the evaluation of the models in the case of MRLs. Firstly,  we by no means argue that EM and F1 are useless, they are the standard metric, and hence we report these results. On top of that, in MRLs some span differences are more nuances and hence we provide an additional, more sensitive, metric. All in all our evaluation shows that EM and F1 have a tendency to be strict since they cannot attend to in-token nuances, and hence should be treated as a lower bound, which is also useful for model analysis.
> Secondly, our metrics for evaluating the metrics itself, are based on good and bad spans collected from humans, and in that sense, it is effectively human-based. As we describe in line 454e we have two types of measures for evaluating the metric: positive (where we expect the metric to give high scores to correct spans) and one is negative (where we expect the metric to give low scores to incorrect spans). Both evaluations rely on manually collected gold data of good and bad spans, and we argue that this is a methodologically solid way to create a robust evaluation for the new metric.  We are open to adding additional forms of human-based evaluation for our TLNLS to the camera-ready – could you please elaborate on what kind of human judgments you would like to see on top of the human tagging of good/bad spans as we described above?
>
>
> Q: In 3.1 (Annotation Philosophy), you mention that you increased diversification in three ways. The first was clear: include different structures and different topics. The other two methods just say that you made sure the dataset was “diverse”—what metrics are you using for diversity?
>
> A: The second endeavor was to select paragraphs from different and diverse topics in wikipedia. The third endeavor was to systematically look for repeating patterns that were used by the annotators during the evaluation of the annotators. and instruct them to create different kinds of questions. We will explain this more clearly in the camera-ready version.
>
> Q: How did you create the multiple answer spans to augment the train and test samples?
>
> A: The test and validation were validated completely by the author of the paper. When we validate them we manually add all the relevant spans beside the one chosen by the annotator. We will clarify this in the camera ready version.
>
> Q:In line 351, what is a “correct question”?
>
> A: “Correct questions” are defined in the paper to be questions that didn’t fall into any one of the “incorrect question” categories. The categories of “incorrect questions” are as follows: 1. Questions that were marked “unanswerable” but were actually answerable, i.e., their answer appeared in the text; 2. “Answerable”-marked questions that their answer did not appear in the text (the marked span was wrong); 3. Questions that were judged ungrammatical. The rest of the questions are judged “correct”.
> Q: At the beginning of 4.3, do you just mean “pairwise?”
>
> A: Using “pairwise” is a lot more concise and clear. We will use that.
>
> Q: In line 452, did you mean if the number of digits in either spam is more than half the total number of digits in each span?
>
> A: No. We mean that out of all the characters in a single span, at least half are digits (we evaluate this separately for each span). For example in the string “asd23” two out of five are digits. This is used mainly to filter out dates that are expressed only as digits and punctuation which is the only scenario that we found TLNLS to be underperforming.
>
> Regarding typos - we will be sure to proofread the entire paper and correct all remaining typos.

---

### Official Review · Reviewer_cQ9w · 2023-08-05

**Soundness:** 4

**Excitement:**

3: Ambivalent: It has merits (e.g., it reports state-of-the-art results, the idea is nice), but there are key weaknesses (e.g., it describes incremental work), and it can significantly benefit from another round of revision. However, I won't object to accepting it if my co-reviewers champion it.

**Paper Topic And Main Contributions:**

The paper presents a new dataset for MRC in Hebrew, built based on a rigorous data collection and filtering process, using paragraphs from Wikipedia and a newspaper with technical news. Because Hebrew is a morphologically rich language, to allow for such variations in the potential answers, a new string matching method is proposed, that does not penalize morphologically inflected words.

**Questions For The Authors:**

Explain how the answers were annotated.

What are the statistics of the final dataset?

What kind of errors that ParaShoot makes does HeQ help mitigate?

**Reasons To Accept:**

The dataset constructed, and the process used to collect it, are useful resources for NLP, and also further emphasize the different processing requirements of different languages. The issues surrounding morphologically rich languages are well presented and help justify the necessity of the dataset and the new string matching process.

**Reasons To Reject:**

The main result of the work presented in the paper is the dataset, but there are insufficient information regarding it:

- the process of obtaining questions is described, but what are the answers? are they developed in parallel?

- what are the final statistics of the dataset? Section 3.2 provides details of data obtained based on Wikipedia, but no details about the data obtained from Geektime.

- the authors argue for the diversity of the data gathered, but there is no quantification, and this is not difficult to provide. Even a shallow types vs. token statistic would be useful.

- it seems like a good idea to choose negative spans from the wrongly predicted answers, but they may not necessarily be good for evaluating the string matching method. How about partly overlapping spans with the positives? Or other such spans that are more difficult negatives? The purpose of the new string match method is to deal correctly with morphological variations, including misleading ones. It would also be useful to have some error analysis for the TLNLS. What situations does it get wrong?

- some discussion and error analysis on the new dataset would be useful. Contrasting this to ParaShoot would also be useful, to show the benefits of using the new dataset (e.g. examples of ParaShoot that the model trained on HeQ gets right, to illustrate what phenomena/situations the new dataset helps model better)

- the ParaShoot dataset is frequently referenced but not described, and not well contrasted with the dataset presented here.

**Reproducibility:**

4: Could mostly reproduce the results, but there may be some variation because of sample variance or minor variations in their interpretation of the protocol or method.

**Reviewer Confidence:**

2: Willing to defend my evaluation, but it is fairly likely that I missed some details, didn't understand some central points, or can't be sure about the novelty of the work.

---

> ### Author Rebuttal · Authors · 2023-08-28
>
> Thank you for the detailed review! We were happy to read that you also identified the need for a large-scale MRL dataset.
>
> Q: “the process of obtaining questions is described, but what are the answers?“
>
> A: The questions and answers are written in tandem. We report this in lines 292-293
>
> Q: What are the final statistics of the dataset? Section 3.2 provides details of data obtained based on Wikipedia, but no details about the data obtained from Geektime.
>
> A: The overall dataset contains 30147 samples. out of which 15025 are from Wikipedia and 15122 are from Geektime. We did not write the size of each split, but we will be sure to add it to the camera ready once it gets accepted.
>
> Q: the authors argue for the diversity of the data gathered, but there is no quantification, and this is not difficult to provide. Even a shallow types vs. token statistic would be useful.
>
> A: We will add token uniqueness between the two domains and between intra-domain categories. If there are more diversion metrics that you see as beneficial, we’d be happy to add them too!
>
> Q: it seems like a good idea to choose negative spans from the wrongly predicted answers, but they may not necessarily be good for evaluating the string matching method. How about partly overlapping spans with the positives? Or other such spans that are more difficult negatives? The purpose of the new string match method is to deal correctly with morphological variations, including misleading ones. It would also be useful to have some error analysis for the TLNLS. What situations does it get wrong?
>
> A: The purpose of TLNLS is to be a good discriminator between two models and not between two random spans (we will clarify this point in the paper). For this reason, we believe that the best way to evaluate it is by using the actual predictions of models. While we believe this is the best way to evaluate the model, we agree that adding a partly overlapping span will strengthen the analysis of the new metric, and we will add this to the camera ready. Regarding error analysis, we believe that the negative analysis does exactly that - it evaluates how the metric scores the model when it predicts an incorrect span.  If you have further ideas for evaluation, we will be happy to include them.
>
> Q: Some discussion and error analysis on the new dataset would be useful.
>
> A: We scrutinized several aspects of the dataset (different sizes, different domains) as an analysis. We have additional error analysis that was dropped due to lack of space (In short, the model fails much more on samples that require inference, more so than those quoting the answer verbatim). We will make sure to add this analysis to the camera-ready version.
>
> Q: Contrasting this to ParaShoot would also be useful, to show the benefits of using the new dataset (e.g. examples of ParaShoot that the model trained on HeQ gets right, to illustrate what phenomena/situations the new dataset helps model better)
>
> A: We include a direct comparison between HeQ and Parashoot in the section that starts in line 590. Note that Parashoot is a very small dataset (having 1792 samples for training). This results in very simple models that use relatively shallow heuristics. For example, they are easily misled to select an answer if there is a relevant type in the text (for example, a person, when the question is “who”). The models created based on the much larger  HeQ are more nuanced in this case.  We didn't include a direct comparison because we didn’t see it as a significant contribution to our analysis - we argue that the high-quality data and the fact that new Hebrew models do not show an improvement over older ones when semantic tasks like MRC are involved is the heart of the paper. Nevertheless, we will add this to the camera ready.
>
> Q: the ParaShoot dataset is frequently referenced but not described, and not well contrasted with the dataset presented here.
>
> A: We discuss Parashoot in several places in the paper, starting with line 053. Mostly we discuss the related work section starting line 653. We will add a more elaborate description of ParaShoot, including the dataset statistics, as well as a direct comparison to it, to the camera-ready version
>
> Regarding reproducibility - our data is in the same format as SQuAD, and can be easily trained with the available SQuAD scripts. In addition, we will release all the data and trained models necessary to replicate our results. If there are additional aspects that we missed and you wish to be disclosed to allow for easy reproducibility, please let is know. We’d very much like to improve on that.

---

### Meta-Review · Area_Chair_Wm9s · 2023-09-13

**Recommendation:** 4

**Metareview:**

This paper introduces a machine reading comprehension dataset for Hebrew, HeQ, consisting of approximately 30K question-answer pairs, used to demonstrate comprehension. It likewise details the collection and annotation pipelines, a revised evaluation metric (TLNLS, which is better suited for the complex morphological patterns exhibited by Hebrew), benchmarking experiments, as well as comparisons to an existing machine reading comprehension dataset ParaShoot. Reviewers praised the paper’s clarity and motivation, its literature review, and the fact that it presents a high-quality resource with few to no viable alternatives. They did raise individual concerns in their reviews, however (e.g. extensibility of the proposed evaluation metric TLNLS to other morphologically rich languages and lack of human evaluation regarding TLNLS). Reviewer cQ9w raised several additional concerns, commenting the paper would benefit from (1) added details pertaining to the dataset, e.g. human evaluation, statistics, evaluation of diversity, (2) more detailed comparison to Parashoot, and (3) detailed error analysis of the models trained on HeQ. In their rebuttals, the authors addressed all of these concerns extensively, sharing the quantitative and qualitative results of their error analysis and empirical results from their comparisons between Parashoot and HeQ.

To summarize, this paper calls for a fair number of revisions, but much of the work required for these revisions is already complete, as demonstrated in the author rebuttals. What remains is to incorporate this work in writing and ensure that all reviewers comments and concerns have been addressed.

---

### Decision · Program_Chairs · 2023-10-07

**Decision:**

Accept-Findings

**Comment:**

This paper introduces a machine reading comprehension dataset for Hebrew, HeQ, consisting of approximately 30K question-answer pairs, used to demonstrate comprehension. It likewise details the collection and annotation pipelines, a revised evaluation metric (TLNLS, which is better suited for the complex morphological patterns exhibited by Hebrew), benchmarking experiments, as well as comparisons to an existing machine reading comprehension dataset ParaShoot. Reviewers praised the paper’s clarity and motivation, its literature review, and the fact that it presents a high-quality resource with few to no viable alternatives. They did raise individual concerns in their reviews, however (e.g. extensibility of the proposed evaluation metric TLNLS to other morphologically rich languages and lack of human evaluation regarding TLNLS). Reviewer cQ9w raised several additional concerns, commenting the paper would benefit from (1) added details pertaining to the dataset, e.g. human evaluation, statistics, evaluation of diversity, (2) more detailed comparison to Parashoot, and (3) detailed error analysis of the models trained on HeQ. In their rebuttals, the authors addressed all of these concerns extensively, sharing the quantitative and qualitative results of their error analysis and empirical results from their comparisons between Parashoot and HeQ.

To summarize, this paper calls for a fair number of revisions, but much of the work required for these revisions is already complete, as demonstrated in the author rebuttals. What remains is to incorporate this work in writing and ensure that all reviewers comments and concerns have been addressed.